# A host driven parasitoid syndrome: Convergent evolution of multiple traits associated with woodboring hosts in Ichneumonidae (Hymenoptera, Ichneumonoidea)

**Davide Dal Pos** *, **Barbara J. Sharanowski**

Department of Biology, University of Central Florida, Orlando, Florida, United States of America

* daveliga@gmail.com, davide.dalpos@ucf.edu

## Abstract

The evolution of convergent phenotypes is of major interest in biology because of their omni-presence and ability to inform the study of evolutionary novelty and constraint. Convergent phenotypes can be combinations of traits that evolve concertedly, called syndromes, and these can be shaped by a common environmental pressure. Parasitoid wasps which use a wide variety of arthropod hosts have also repeatedly and convergently switched host use across their evolutionary history. They thus represent a natural laboratory for the evolution of trait syndromes that are associated with parasitism of specific hosts and host substrates. In this study, we tested the evolution of co-evolving characters in the highly diverse family Ichneumonidae associated with ovipositing in a specific and well-defined substrate: wood. Using a newly constructed phylogeny and an existing morphological dataset, we identified six traits correlated with the wood-boring lifestyle that demonstrate convergent evolution. At least one trait, the presence of teeth on the ovipositor, typically preceded the evolution of other traits and possibly the switch to parasitism of wood-boring hosts. For each trait, we provide a historical review of their associations with wood-boring parasitoids, reevaluate the function of some characters, and suggest future coding improvements. Overall, we demonstrate the convergent evolution of multiple traits associated with parasitism of woodboring hosts and propose a syndrome in a hyper diverse lineage of parasitoid wasps.

## Introduction

Convergent evolution is a cornerstone concept in evolutionary biology for several compelling reasons. As articulated by Stayton [1] its significance is underscored by two primary factors: (1) its pervasive occurrence throughout the history of life on Earth; and (2) its interpretation as a predictable consequence of evolution driven by natural selection. Phenotypic convergence provides visual evidence of the power of natural selection, particularly when considering

---

**Data Availability Statement:** All relevant data are within the manuscript and its Supporting Information files.

**Funding:** This research was supported by the US National Science Foundation (NSF: DEB-1916788)

---

grant awarded to Barbara J. Sharanowski. Article processing charges were provided by the UCF College of Graduate Studies Open Access Publishing Fund to Davide Dal Pos. The sponsors did not play a role in the study designa, writing or data collection.

**Competing interests:** The authors have declared that no competing interests exist

adaptations due to shared environmental pressures [2]. More practically, convergent evolution serves as a valuable tool for biologists, providing a natural laboratory for repeated experiments in evolution and providing researchers with the replicated events needed for statistical power [1–5]. The study of convergent evolution is broad, encompassing a wide range of taxa and includes examinations of morphology (such as phenotypic convergence) and behavior, as well as investigations across different timescales [6–8].

Definitions of convergent evolution have varied depending on whether or not adaptation is invoked or whether a process for convergence is defined, such as the use of similar developmental or genetic pathways to achieve the same phenotype [2,3,9,10]. Here we employ the definition given by Losos [3], which defines convergence as the "*independent evolution of similar features in different evolutionary lineages.*" Characters can be deemed "similar" if they share similarities in their phenotype, independently or dependently from the genetic underpinning. This aspect requires careful morphological assessment, because what may appear near identical externally could exhibit significant differences in skeletal-musculature organization [11,12]. This is especially true for insects where external morphology is far more frequently used than internal morphology. Examples range from the eyes of Siphonaptera, which are actually ocelli [13], to the increase of leg dimensions in independent insect lineages, which represent a complex evolution involving multiple leg expansions [14]. This morphological similarity is primarily determined by comparative morphologists, who propose homology statements of characters if they occupy the same area (topology) or if the relations with other sclerites and/or muscles are maintained (connectivity) in multiple taxa [15]. Vogt et al. [16] termed this sameness *structural equivalence*, differentiating it from the classical concept of homology, which usually requires a phylogenetic context (similarity due to common ancestry) [17].

Characters could also be considered "similar" if their function is the same, even though they have a different genetic pathway and morphological organization, as natural selection acts upon the functional consequences of traits, rather than the traits themselves [e.g., 3,18]. In parasitoid wasps, for example, unrelated taxa have different mechanisms for bracing the ovipositor during oviposition into a wood-boring substrate. In Labeninae there are modifications of the coxae, whereas members of Rhyssinae have ovipositor guides [19]. Here, we focus on structural rather than functional equivalence to determine similarity.

When convergence involves the co-appearance of multiple traits, it is often referred to as a *syndrome*, defined as multiple traits that evolved concurrently in response to a common environmental pressure [20–23]. Syndromes remain poorly studied, with various factors that may contribute to their formation. Possible mechanisms for multi-trait convergence include: supergenes created by chromosomal inversions that link multiple genetic elements; genetic linkage due to chromosomal proximity; and/or pleiotropic effects of single genes [e.g., 20,24–26]. Interesting is the role of evolutionary precursors in trait formation–ancestral traits or states that potentially facilitate the convergent evolution of related traits (often referred to as positive constraints)–which could set the stage for parallel evolution in related taxa [e.g., 27,28].

Hyperdiverse taxa, like parasitoid wasps, present an intriguing opportunity for testing convergent evolution and syndromes. The notable rate of convergence observed across diverse lineages [29] provides a substantial number of replications. Parasitoid wasps live off arthropod hosts, mainly insects, by laying eggs in or on hosts, subsequently developing off the host's tissues until its death, and emerging as an adult after pupation to complete the life cycle. As parasitoid wasps rely on hosts to complete their lifecycle, the type of host drives the evolution of several morphological traits related to finding, ovipositing, developing on, and ultimately exiting from the host [30]. Hosts may be exposed or may be more hidden within a substrate. The

types of substrates are numerous, ranging from various plant tissues [31,32] to insects eggs [33,34] and spider sacs [35,36]. The host substrate can present a formidable obstacle that the wasp must overcome to reach its host [37] and thus likely plays a pivotal role in morphological evolution. Thus, for parasitoids, the evolution of several morphological traits is host-driven and thus, specific traits should be tightly correlated with specific hosts or host substrates.

Ichneumonidae is an especially ideal parasitoid wasp lineage for examining convergent evolution and trait syndromes because of their extensive species diversity, varied life history strategies, including diverse hosts, and convergent adaptations related to host use [29,30,38–41]. With more than 25,000 described species, Ichneumonidae is considered one of the largest families of Hymenoptera [42,43]. The remarkable species diversity is paralleled by the numerous and diverse parasitism strategies, showcasing a broad spectrum of host specificity that spans from holometabolous insects to spiders [30,44,45]. Additionally, the variety of substrates utilized by their hosts is extensive and there have been repeated host shifts across the phylogeny [30,32].

Parasitism of hosts within a woody substrate has been shown to be ancestral for Ichneumonidae [32,42] but also for the evolution of parasitism itself within Hymenoptera [19,46]. Hosts use the wood for food, but also for concealment and protection against predators and environmental extremes [47]. In response to the challenges of wood, parasitoid wasps have developed specific characters associated with both ovipositing into and emerging from wood, some of which have been referred to as adaptions given the impact these traits have on successful parasitism and thus survival [19]. Wasps that want to utilize hosts found in these substrates have three main obstacles that must be overcome: 1) locating the host inside the wood, 2) ovipositing within or onto the host; and 3) emerging from the substrate after larval development [19]. The life cycle of a woodboring Ichneumonid parasitoid, *Megarhyssa atrata* (Fabricius, 1781) is provided in Fig 1 as an example.

It is important to differentiate between subcortical parasitoids, which target hosts located under bark (e.g., the genus *Rhimphoctona* Förster, Ichneumonidae: Campopleginae) [48], and woodboring parasitoids, which target specimens deeply concealed within lignified plant tissue (e.g., Ichneumonidae: Rhyssinae). Here we focus only on wood-boring parasitoids, as subcortical parasitoids may not face the same challenges to locate and reach their hosts and thus may not have the same distinctive morphologies commonly associated with wood-boring parasitoids, as identified by previous authors [e.g., 19,49].

Here we look at traits associated with parasitism of wood-boring hosts across the entire family of Ichneumonidae to: (1) test for traits correlated with having a host within a woody substrate; (2) determine if these traits co-evolved, thereby forming a syndrome; and (3) to analyze and discuss the syndrome itself through careful examination of the traits. To complete these objectives, we utilized morphological characters from Bennett et al. [32] as this study contains the most comprehensive morphological dataset to date for Ichneumonidae, covering almost all major lineages. Based on the literature, we selected traits from this dataset that are putatively associated with utilizing wood-boring hosts. Next, we constructed a chimeric phylogeny by combining the taxon-rich dataset from Bennett et al. [32] and the gene-rich dataset from Sharanowski et al. [42]. We then tested for phylogenetic correlation for select traits and performed ancestral state reconstructions (ASR) to examine the relative timing of trait evolution. Finally, to facilitate further study on this trait syndrome, we provide (1) a thorough review of each correlated character in light of our results; (2) critical discussions on the limitations associated with each character; (3) considerations for enhancing the coding of these characters in future studies; and (4) alignment of the terminology with the Hymenoptera Anatomy Ontology (HAO) [50].

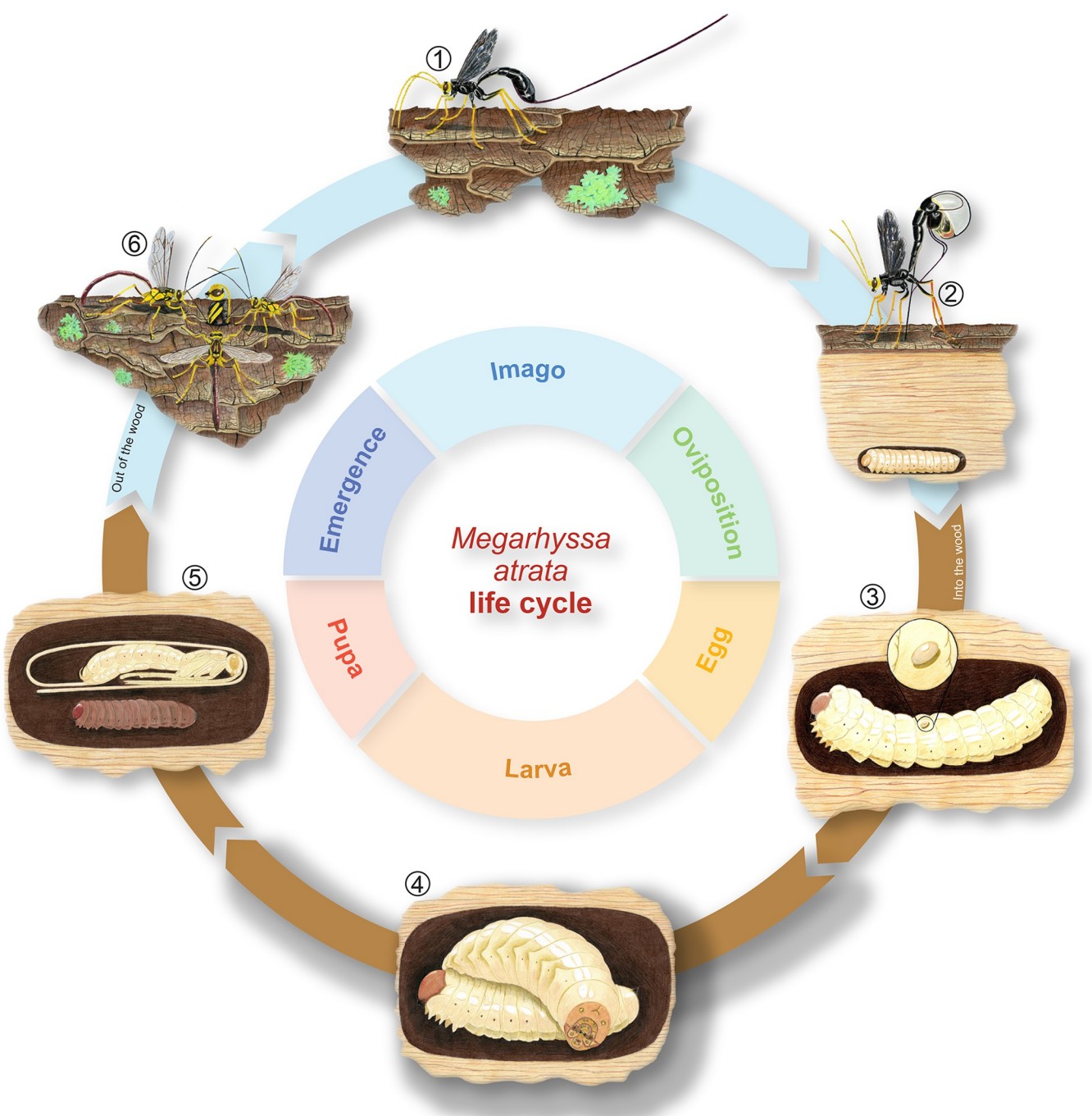

**Fig 1. Life cycle of the woodboring idiobiont ectoparasitoid *Megarhyssa atrata* (Fabricius, 1781).** (1) female wasp locating potential host by tapping the antennae; (2) wasp ovipositing within wood trying to reach the deeply concealed larvae; (3) host larvae with parasitoid egg attached; (4) parasitoid larvae fully developed (on top) and feeding on the host larvae (underneath); (5) pupal stage of the parasitoid wasp (on top) and host larvae dead (below); (6) emergence of the newly hatched parasitoid wasp adult with males waiting to mate.

## Material and methods

### The characters

We selected 20 characters from the141 coded by Bennett et al. [32] for analysis. The first character is the oviposition substrate itself, as in lignified tissue or not (Character 1, S1 File). Then, we conducted a thorough review of the literature and identified 10 of Bennett et al.'s [32] characters that have been historically associated with a woodboring lifestyle (see Table 1 and characters 2–10, and 18, S1 File). Then, we chose eight additional characters not previously documented but potentially relevant for ovipositing into wood, based on their overall shape and body position. A rationale for these choices is provided in S1 File (characters 11–17, and 19). This approach was based on the premise that not all traits related to a wood-boring lifestyle may have been previously identified. All other characters from Bennett et al.'s [32] study were clearly unrelated to oviposition or emergence from wood (e.g., all larval characters) and were thus excluded.

Although we tested 20 characters from Bennett et al. [32], only eight were significantly correlated with parasitism of wood-boring hosts (see results). Detailed descriptions of all 20 characters and the coding modifications we made can be found in S1 Table. Terminological alignment of the characters with the Hymenoptera Anatomy Ontology [50] can be found in S1 Table.

To facilitate understanding, we introduce here the eight correlated characters and the woodboring substrate. We have provided a number for both the substrate and each character, corresponding to its reference number used throughout the paper. The statements in square brackets indicate how the character is alternatively referred to in S1 Table. For each of these characters, we also provided the modified coding (from Bennett et al. [32]) that we employed for our analyses.

**1-Woodboring substrate [Host substrate, type; Substrate, woodboring].**   Several Hymenopteran morphological characters have been associated with the woodboring substrate, either living in wood as immatures or parasitizing hosts living in wood. We provide a complete list of these characters across Hymenoptera accompanied by their proposed functions and a list of references associated with them (Table 1). For this study focusing on Ichneumonidae, we used Bennett et al.'s [32] coding for "*Oviposition location*" (character 140) as follow: (0) lignified plant tissue; (1) other substrates.

**2–First valvula (1vv) with teeth [1vv, teeth].**   The presence of teeth (serration) on the apical tip of the terebra are integral to the mechanical aspects of wood penetration [19,30,56,68,71,87]. The teeth facilitate the drilling process by either rasping or breaking the wood fibers to either reach the host or lay free living wasp larvae (e.g., wood wasps) [88]. We used Bennett et al.'s [32] coding for "*Ovipositor ventral valve*" (character 97) as follows: (0) with teeth apically; (1) without teeth apically.

**3–Elongated terebra [Terebra, length].**   In order to access hosts deeply concealed within wood, parasitoid Hymenoptera tend to develop a longer terebra compared to their non-wood-boring counterparts [19,72]. Lengths of terebra can vary, exceeding eight to nine times the body size in some taxa, like in Megalyridae and some Ichneumonoidea [19,56,72,89]. We used Bennett et al.'s [32] coding for "*Ovipositor length*" (character 96) as follows: (0) Terebra shorter than the metasoma, and (1) terebra longer than the metasoma.

**4–Modified ventral margin of clypeus [Ventral clypeal margin, shape; Clypeus, modification].**   Modifications of the apical margin of the clypeus have been historically correlated with emergence from wood [19]. According to Turrisi and Vilhelmsen [47], Aulacidae employs tooth-like processes on the medio-apical margin of clypeus to facilitate the disintegration of the surface during emergence from wood. Similar structures with analogous functions

**Table 1. List of the characters associated with the wood-boring lifestyle in Hymenoptera (some part of the lifecycle in wood, either as wood wasps or as parasitoids of wood-boring hosts).**

| Body part | Location | Character | Function | References |
|---|---|---|---|---|
| **Head** | Antennae | Multiporous plate sensillae | Collection chemical cue | Basibuyuk and Quicke [51], Broad and Quicke [52] |
| **Head** | Antennae | *Hammer-like distal flagellomere | Generating vibrational sound | Broad and Quicke [52], Vilhelmsen et al. [53] |
| **Head** | Clypeus | *Clypeus with a median tooth-like process | Crumbling and removing debris | Turrisi and Vilhelmsen [47], Turrisi et al. [54], Quicke [55] |
| **Head** | Clypeus | Concave clypeus (Cyclostome condition) | Allowing a wider range of movement of the mandibles | Belokobylskij [56] |
| **Head** | Labrum | Labrum upcurved with an anteroventral brush of setae | Functioning as a broom to sweep dust and debris away from the mandibles | Vilhelmsen [57] |
| **Head** | Mandible | Presence of zinc | Reinforcement of the teeth for chewing during emergence | Quicke et al. [58] |
| **Head** | Mandible | Baso-lateral mandibular groove | Lateral movability of mandibles; extension of the subantennal groove | Turrisi and Vilhelmsen [47] |
| **Head** | Occiput | Very broad occipital carina | Prevent debris from fouling the back of the head | Turrisi et al. [54] |
| **Head** | Vertex | Ocellar corona; parascrobal crests | Anchor the head while the mandibles are chewing to exit the wood chamber; remove debris from the galleries; drag the wasp through the galleries | Turrisi and Vilhelmsen [47], Engel and Grimaldi [59], Gibson [60], Krogman and Burks [61], LaSalle and Stage [62] |
| **Head** | Vertex | Subantennal grooves; supra-antennal grooves (not always correlated with wood in Ichneumonoidea) | Accommodate antennal base, to protect them during emergence | Turrisi and Vilhelmsen [47], Turrisi et al. [54], Vilhelmsen [63], Vilhelmsen et al. [64], Vilhelmsen [65] |
| **Mesosoma** | Fore leg | Apical tibial spur (calcar) with comb and a notched basitarsus | Antennal cleaning after emergence | Basibuyuk and Quicke [66] |
| **Mesosoma** | Fore tibia | Enlarged and containing subgenual organ | Pick up vibrational sounds and transduce them into nerve impulses | Vilhelmsen et al. [53], Vilhelmsen et al. [67] |
| **Mesosoma** | Hind coxa | Presence of a groove | Grip of the terebra during drilling | Gauld and Wahl [68] |
| **Mesosoma** | Hind leg | Apical margin of hind tibia with setae arranged in a spatula | Wing cleaning | Basibuyuk and Quicke [69], Vilhelmsen et al. [70] |
| **Mesosoma** | Mesoscutellum | Parascutal lobe; supra-tegular tooth-like process (close to the tegula) | Cover and protect the fore wing base from abrasion | Turrisi & Vilhelmsen (2010) |
| **Mesosoma** | Mesoscutum | *Transverse sculpture; transverse anterior projection | Facilitate removing debris; bracing the thorax | Vilhelmsen and Turrisi [19], Turrisi and Vilhelmsen [47], Quicke [55], Krogman and Burks [61], Gauld and Wahl [68] |
| **Metasoma** | 2nd valvula | *Serration or teeth-like processes | Cut substrate (mostly wood) | Vincent and King [71] |
| **Metasoma** | 2nd valvulae | *Enclosing first valvulae | Stabilizing the ovipositor during drilling | Santos and Perrard [23] |
| **Metasoma** | Ovipositor | *Long terebra | Reach host deeply concealed in substrate (usually wood) | Vilhelmsen and Turrisi [19], Nénon et al. [72] |
| **Metasoma** | Ovipositor | Internalization of terebra, either entirely (e.g., Orussidae) or during oviposition (e.g., Rhyssinae) | Facilitate the carrying of a long ovipositor | Vilhelmsen [49], Vilhelmsen et al. [53], Vilhelmsen et al. [67], Le Lannic and Nénon [73], Sivinski and Aluja [74] |
| **Metasoma** | Ovipositor | Transverse striation on ovipositor sheaths (flexibility) | Anchoring the tip of the terebra in the initial phase of ovipositing | Vilhelmsen [75], Rodd [76] |
| **Metasoma** | Ovipositor | Steering mechanism | Steering the terebra during probing | Eggs et al. [37], Quicke and Fitton [77], Quicke et al. [78], Spradbery [79], Quicke [80], Quicke and Marsh [81] |
| **Metasoma** | Sternites | *Ovipositor guides | Handling a very long ovipositor during oviposition | Vilhelmsen and Turrisi [19], Vincent and King [71], Gardiner [82] |
| **Metasoma** | Terebra | Presence of zinc or manganese or calcium | Reinforcement of the tip for drilling | Quicke et al. [58], Vincent and King [71], Quicke et al. [83] |
| **Metasoma** | Terebra | Sensillae and secretory structures | Lubricate the ovipositor and possibly degrade the wood during drilling | Nénon et al. [72], Nénon et al. [84] |
| **Metasoma** | Terebra | Cross-section | Minimize friction & maximize internal lumen of the passage of the egg | Vilhelmsen and Turrisi [19], Quicke et al. [85], Cooper [86] |

*(Continued)*

**Table 1.** (Continued)

| Body part | Location | Character | Function | References |
|---|---|---|---|---|
| **Metasoma** | Tergites | *Elongated abdominal tergum 9 | Hosting stronger ovipositional muscle | Santos and Perrard [23] |
| **Metasoma** | Valvulae | Thick cuticle | Reinforcement of the terebra during drilling | Quicke et al. [85] |

**Body part** = Specific tagma on insects; **Location** = Location of the character within the associated body part; **Function** = Proposed functionality for the character from the literature; **References** = List of references that propose a specific association of the character with woodboring lifestyle and/or function of the character

* = Identify the characters tested in our study.

have been documented in other insect families, including Stephanidae and some Ichneumonidae that pupate within wood [55] (Table 1). We used Bennett et al.'s [32] coding for "*Clypeal margin in anterior view*" (character 3) as follows: (0) simple, truncate to slightly concave; (1) modified, either bilobed or with a median denticles (or both).

**5–Elongated abdominal tergum 9 [Abdominal tergum 9, elongation].** The enlargement of the apical tergites in Ichneumonidae, such as the abdominal tergum 9, has been correlated with hard substrate penetration [55]. Santos and Perrard [23] used this enlargement as a proxy for augmented oviposition muscles with the presumed function that these enlarged muscles facilitate the penetration of hard substrates, such as wood or mud. We used Bennett et al.'s [32] coding for "*Apical segment of female metasoma*" (character 92) as follows: (0) short, not elongated; (1) elongated, with or without horn or bosses.

**6–Modified apical flagellomere [Apical flagellomere, shape].** Some parasitoid Hymenoptera have an antennal modification of the distal portion of the apical segment to be distally flat (hammer-like) with a surface that does not bear any hairs or sensillae [52: Fig 1]. This modification enables the direct creation of vibrational sounds by tapping the substrate, and thereby facilitating the detection of the host within the woody substrate [19,52]. Subsequently, these vibrations are collected by the subgenual organ located in the hind legs, functioning as a hearing device for the wasps [19,47,52,53]. Within Ichneumonidae, the adaptive significance of vibrational sounds associated with a deeply concealed host has been recently analyzed via phylogenetic comparative analyses [52]. We used Bennett et al.'s [32] coding for "*Apical flagellomere of female*" (character 7) as follows: (0) simple, not flattened; (1) flattened.

**7–Rugulose mesoscutum [Mesoscutum, dorsal surface].** A mesoscutum with a strong rugulose dorsal sculpture has been associated with the wood-boring lifestyle [19,30,47,55], and has been recorded in several Hymenopteran taxa, mainly Ibaliidae (Cynipoidea) [90], Ichneumonoidea [e.g., 44,91], and Chalcidoidea [e.g., 61] (Table 1). Quicke [30] listed this trait as one of the convergent features in parasitoid wasps. Turrisi and Vilhelmsen [47] suggested that these pronounced sculptures may serve different purposes: providing structural support to the body, aiding in the removal of debris during wood penetration, and protecting delicate structures, such as the proximal section of the wings. Additionally, Quicke [55] postulated that the rugulose mesoscutum plays a crucial role in securing a grip on the sides of its burrow, facilitating the wasp's escape from the substrate. We used Bennett et al.'s [32] coding for "*Mesoscutum*" (character 21) as follows: (0) smooth; (1) with transverse rugae.

**8–Ovipositor guides [Ovipositor guides].** Handling and maneuvering an extended ovipositor poses a considerable challenge, causing various taxa to evolve external supports for vertically orienting the terebra [19]. Within Ichneumonidae, the subfamily Rhyssinae employs a distinctive mechanism known as ovipositor guides, which involves a median groove in the sternal region of the metasoma, paired by clips on multiple sterna, in which the terebras runs during the oviposition process, securing it close to the metasoma and thereby enhancing the

overall stability of the process [19,71,82]. We used Bennett et al.'s [32] coding for "*Posterior sternites of females*" (character 93) as follows: (0) absent; (1) present.

**9–First valvula (1vv) enclosing second valvula (2vv) [1vv, enclosing 2vv].**   One intriguing modification of the terebra is the dorsal expansion of the first valvula, forming a distinctive lobe that envelops the second valvula [e.g., 92,93]. Although the literature provides limited (if any) insights into the functionality of this lobe, Santos and Perrard [23] treated it as a possible mechanism to stabilize the ovipositor while probing into wood. We used Bennett et al.'s [32] coding for "*Ovipositor ventral valve*" (character 98) as follows: (0) not enclosing 2vv; (1) enclosing 2vv.

## Phylogenetic analyses

To obtain a robust phylogeny with good taxonomic sampling, we combined the morphological and molecular data from Bennett et al. [32] and the amino acid data from Sharanowski et al. [42]. Bennett et al.'s [32] data included 141 morphological characters for 134 taxa and 1,309 nucleotides from the following genes (*28S* rDNA, *COI* mtDNA, and the protein-coding gene *EF1-alpha*). Sharanowski et al.'s [42] data included 50,145 amino acids characters from 541 genes derived from an anchored hybrid enrichment approach using the Hymenoptera probe set [94]. To combine the datasets, we integrated the data of shared species between these two datasets. In cases where we couldn't find a species-specific match, we proceeded to integrate data at the genus level. Two notable exceptions to this procedure were: 1) the integration of an unidentified Ichneumonini (referred as "Ichneumonini_1") from Sharanowski et al. [42] with *Coelichneumon eximius* in Bennett et al. [32]; and 2) the association of the AHE data of *Rhyssalus* sp. from Sharanowski et al. [42] with *Doryctes erythromelas* in Bennett et al. [32] because we wanted both outgroups to have data for all genes. See S2 Table for a complete list of generated chimeras across taxa.

The resulting dataset was analyzed using IQ-Tree v.2.2.2.7 [95] on the CIPRES Science Gateway [96]. The analysis involved a partitioned approach, where the following models were applied based on the respective data types: the MK model for morphology, GTR+G for nucleotide data, and WAG+G for the amino acid data. We also performed 1000 ultra-fast bootstraps to assess nodal support.

When the entire dataset was analyzed, the subfamily Lycorininae was recovered out of the Ichneumonidae, in a polytomy with Braconidae. Long branch attraction was suspected across Lycorininae and a few other taxa, so we conducted a series of long branch exclusion tests [97]. These tests revealed several rogue taxa (with highly variable placement across the tree) whose inclusion impacted other phylogenetic relationships. See S2 File for a more a summary of the exclusion test results. Consequently, we excluded rogue taxa from further analyses, including *Anomalon*, *Brachyscleroma*, *Brachycyrtus*, and *Therion*. All of these taxa did not have a representative in the Sharanowski et al. [42] dataset, and thus their labile placement was probably due to the large amount of missing data for these taxa. Tree annotation was performed on the Interactive Tree Of Life (iTOL) version 5 (available at: https://itol.embl.de) [98] and modified in Adobe illustrator.

## Trait correlations and ancestral state reconstructions

BayesTraits V4.1.1 [99; available from http://www.evolution.rdg.ac.uk/] was used to test for correlated evolution between each of the 20 traits considered. For two discreet traits, two different models were tested for best fit: (1) an independent model, which assumes that the two traits have evolved independently, and therefore a transition from 0 to 1 in the first character is independent of the state of the second character; and (2) a dependent model, which assumes

that the two traits are correlated and the rate of change in the first character is dependent on the state of the second character. The models were evaluated using the MCMC setting, estimating the log marginal likelihood using the stepping stone method [100], with 100 and 1,000 iterations per stone and setting all the priors to an exponential with a mean of 10. Log Bayes Factors (logBF) were used to determine which of the two models better fit the data following Mitchell et al. [101]: logBF = 2–4 as having weak support, logBF = 5–9 as having moderate support, and logBF > 10 as having strong support.

In Bennett et al. [32], *Rhimphoctona* (Campopleginae) was coded as a wood-boring parasitoid. However, the taxon appears to be an outlier, lacking many of the characters included in the wood boring syndrome. According to the literature [e.g., 45,102], members of this genus typically target xylophagous insects (e.g., Cerambycidae). However, as noted by Wahl [48], *Rhimphoctona* seems to preferentially attack host larvae living right under bark (subcortical), possibly through probing rather than drilling. To enhance the precision of our analyses, we conducted a second correlation analysis, recoding the substrate for *Rhimphoctona* as not woodboring ("other substrates") to assess its impact on the analyses.

To better identify a putative syndrome, we wanted to know on which branch each trait arose across the tree and see if multiple traits evolved concurrently and repeatedly. To accomplish this, we performed ancestral state reconstructions (ASR) in Mesquite v3.81 [103] on each trait that had a moderate or strong correlation with the woodboring substrate (nine characters in total, including the character for the woodboring substrate). Each character was tested for whether an equal rates (Mk1) or a differential rates model (Asymm2) of character gains and losses best fit the data (S3 Table) using a maximum likelihood reconstruction. Reconstructions of the nine characters can be found in S3 File.

## Results

The resulting phylogenetic tree (Fig 2) is largely consistent with the findings of Bennett et al. [32] and Sharanowski et al. [42] regarding higher-level relationships. The main exception is observed in the placement of Campopleginae, which here is recovered as sister to Ophioninae + Cremastinae. In Sharanowski et al. [42], Ophioninae were identified as sisters to Campopleginae, while in Bennett et al. [32], Cremastinae and Campopleginae formed a polytomy along with Anomaloninae + Ophioninae. These different relationships are not expected to impact the present study, as higher-level Ophioniformes are not known to be wood-boring parasitoids.

Among the 20 characters putatively associated with parasitizing woodboring hosts, only six were strongly correlated (> 10) with the woodboring substrate: *the presence of teeth on the first valvulae, elongated terebra, modified ventral clypeal margin, elongated abdominal tergum 9, rugulose mesoscutum, and 1vv enclosing 2vv* (Table 2). Two other characters (modification of the apical flagellar segment and the presence of ovipositor guides) exhibited a moderate correlation (5–10 LogBF) (Table 2 and 3). All remaining characters tested showed weak or no correlation at all (<5 LogBF). Changing the coding for *Rhimphoctona* yielded only a slight difference in the strength of the correlation for the modified clypeus, which reduced from strong to moderately correlated (Table 2). Importantly, all other characters remained in the same category for strength of correlation.

Examining all other trait-trait correlations, all characters were strongly correlated with at least one other character, with most traits being at least moderately correlated with four or more other traits (Table 3). Character 5, an *elongated abdominal tergum 9*, had at least moderate correlations with six out of the seven other wasp traits, five of which were strongly correlated (Table 3). Character 6, the *modified apical flagellomere* had the least number of

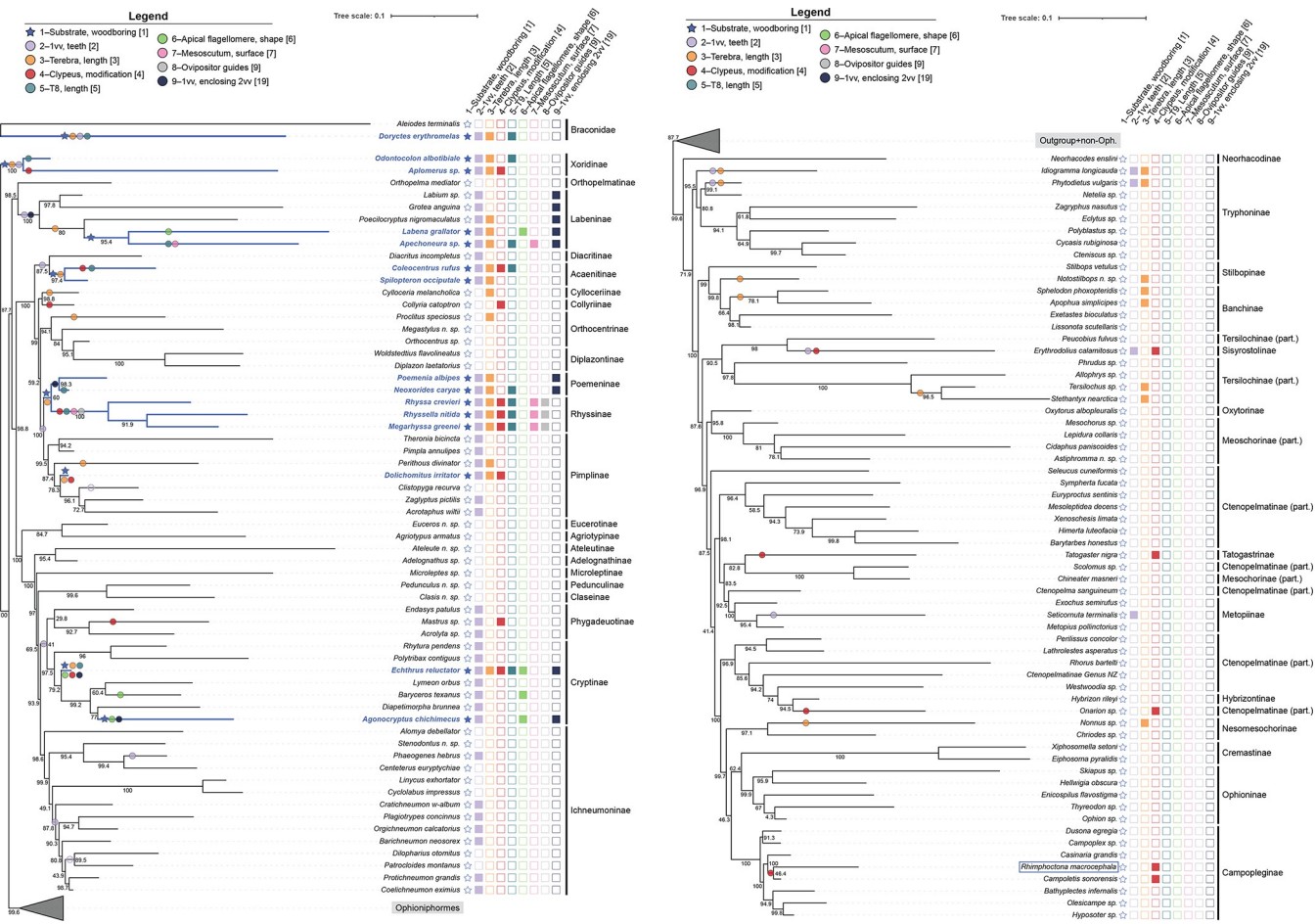

**Fig 2. Maximum-likelihood phylogeny of the subfamily Ichneumonidae based on data from Bennett et al. [32] and Sharanowski et al. [42].** (A) Braconidae (outgroup) and non-Ophioniformes (Ichneumonidae). (B) Ophioniformes (Ichneumonidae). Characters states for each taxon is present beside the taxon name, where filled in squares indicate character presence and empty squares indicate absence. Parasitism in woodboring hosts is presented as a filled in star symbol for easy viewing. Annotation of the characters in circle or star format on the tree highlight where in the phylogeny that character evolved according to the ancestral state reconstructions, where a full circle indicates the appearance of the character and an empty circle indicates a reversal.

correlations to other traits. The Log marginal likelihood for both the more complex model (dependent) and the less complex model (independent), together with the resulting Bayes Factors are reported in S4 Table, including the coding with and without *Rhimphoctona* as woodboring.

## Ancestral state reconstructions

The asymmetrical model (with separate rates for character gains versus losses) was preferred for five characters (wood-boring substrate, teeth on 1vv, long terebra, elongated abdominal tergum 8, and modified apical flagellomere). The equal rates model was preferred for the remaining characters (S3 Table).

For Ichneumonidae, parasitism of wood-boring hosts occurred independently a minimum of 7 times, including Xoridinae. No woodboring parasitoids are present within the Ophioniformes if *Rhimphoctona* is coded as a subcortical parasitoid and thus not woodboring (Fig 2B). Teeth on the first valvula appeared independently 11 times. Four out of five of the appearances occur on a branch preceding the shift to a woodboring host and only one co-occurrence with

**Table 2. Bayes factors from the correlation analyses between all characters and the woodboring lifestyle when *Rhimphoctona* (Campopleginae) was coded as a woodboring and not coded as a woodboring.**

| Characters | *Rhimphoctona* coded as woodboring | *Rhimphoctona* not coded as woodboring |
|---|---|---|
| 2–1vv, teeth [97] | 13.458875* | 13.351593* |
| 3–Terebra, length [96] | 20.932078* | 22.083219* |
| 4–Ventral clypeal margin, shape [3] | **14.524531*** | **8.610323*** |
| 5–Abdominal tergum 9, elongation [92] | 33.289584* | 33.767248* |
| 6–Apical flagellomere, apical margin, shape [7] | 6.137501* | 6.990751* |
| 7–Mesoscutum, dorsal sculpture [21] | 11.086291* | 11.578199* |
| 8–Ovipositor guides [93] | 6.242104* | 5.706375* |
| 9–1vv, enclosing 2vv [97] | 13.871229* | 15.570604* |
| 10–Mandible, shape [5] | -4.894608 | -5.903128 |
| 11–Notaulus, shape [22] | 0.499874 | 0.651516 |
| 12–Flagellum, color [8] | -3.434198 | -4.443347 |
| 13–Genae, shape [12] | 0.596939 | 0.472353 |
| 14–Epomia [20] | -1.729819 | -2.309894 |
| 15–Metathoracic spiracle, shape [34] | 3.622878 | 3.758991 |
| 16–Abdominal tergum 2, shape [79] | 0.69309 | -1.075364 |
| 17–Glymma [78] | 0.375115 | 0.189443 |
| 18–Gastrocoelus [84] | -4.264585 | -5.109145 |
| 19–Abdominal Sternum 8, shape [94] | 1.82945 | 2.277286 |
| 20–Thyridium [82] | -3.110991 | -4.120656 |

* = characters correlated with woodboring; **Bold** = Change of strength of correlation category in the Bayes factor between the two treatments.

the host shift (in Xoridinae). The seven other appearances of the character were not associated with parasitoids of woodboring hosts. Also, a reversal of the character happened twice, one within the Ichneumoninae (*Dilopharius otomitus* + *Patrocloides montanus* clade) and the other within Pimplinae (*Clystopyga recurva*).

An elongated terebra arose independently 13 times. This trait appeared on the same branch as the host shift to woodboring parasitism five times, and on the preceding branch once. All other times this character was not associated with parasitism of wood-boring hosts but were associated with hosts that are deeply concealed (e.g. *P. nigromaculatus* (Labeninae) on the eucalypt gall-forming fly, *Fergusonina flavicornis*). Interestingly, only in one taxon that parasitizes wood boring hosts does a long terebra not occur, *Agonocryptus chichimecus* (Cryptinae).

A modified ventral margin of the clypeus occurred independently a minimum of 11 times, two of which occur in the branch following the host shift to a woodboring host and two times co-occurring on the same branch (in *Dolichomitus irritator*, Pimplinae, and *Echthrus relucta-tor*, Cryptinae). All the other times, the character was not associated with parasitoids of wood-boring hosts. An elongated abdominal tergum nine arose independently a minimum of six times, five of which occurred after the woodboring host shift. This character, when present, always cooccurs with parasitism of woodboring hosts. A modified apical flagellomere appeared independently four times, 2 of which occur on the same branch as the woodboring host shift and once on the following branch. Only one taxon with this character was not associated with a woodboring host.

A rugulose mesoscutum arose independently only two times, all of which occur after the woodboring host shift, while ovipositor guides appeared only once in Ichneumonidae (apomorphy). First valvula enclosing the second valvula occurred independently three times, one occurring after the woodboring host shift (in Poemeninae), one on the same branch (in

**Table 3. Strength of the correlation of wasp traits.**

| # | Character | Correlated Wasp Traits | |
|---|---|---|---|
| | | Strong: Log BF >10 | Moderate: Log BF 5–10 |
| 1 | *Substrate* | (2) *Teeth on 1vv*<br>(3) *Long terebra*<br>(4) *Modified clypeal margin*<br>(5) *Elongated abdominal tergum 9*<br>(7) *Rugulose mesoscutum*<br>(9) *1vv enclosing 2vv* | (6) *Modified apical flagellomere*<br>(8) *Ovipositor guides* |
| 2 | *Teeth on 1vv* | (5) *Elongated abdominal tergum 8*<br>(9) *1vv enclosing 2vv* | (3) *Long terebra*<br>(4) *Modified clypeal margin* |
| 3 | *Long terebra* | (5) *Elongated abdominal tergum 9* | (2) *Teeth on 1vv*<br>(4) *Modified clypeal margin*<br>(7) *Rugulose mesoscutum*<br>(9) *1vv enclosing 2vv* |
| 4 | *Modified clypeal margin* | (5) *Elongated abdominal tergum 8*<br>(9) *1vv enclosing 2vv* | (2) *Teeth on 1vv*<br>(3) *Long terebra*<br>(7) *Rugulose mesoscutum*<br>(8) *Ovipositor guides* |
| 5 | *Elongated abdominal tergum 9* | (2) *Teeth on 1vv*<br>(3) *Long terebra*<br>(4) *Modified clypeal margin*<br>(7) *Rugulose mesoscutum*<br>(8) *Ovipositor guides* | (9) *1vv enclosing 2vv* |
| 6 | *Modified apical flagellomere* | (9) *1vv enclosing 2vv* | |
| 7 | *Rugulose mesoscutum* | (5) *Elongated abdominal tergum 9*<br>(8) *Ovipositor guides* | (3) *Long terebra*<br>(4) *Modified clypeal margin* |
| 8 | *Ovipositor guides* | (5) *Elongated abdominal tergum 9*<br>(7) *Rugulose mesoscutum* | (4) *Modified clypeal margin* |
| 9 | *1vv enclosing 2vv* | (3) *Long terebra*<br>(4) *Modified apical flagellomere* | (4) *Modified clypeal margin*<br>(5) *Elongated abdominal tergum 9* |

A list of the wasp traits at least moderately correlated (Log BF at least 5) with parasitizing hosts within a wood substrate (row 1). All subsequent rows are all trait-trait correlations grouped by their correlation strength according to log Bayes Factors. Character Number (#) refers to the trait number in the coded character matrix, see methods and S1 File.

*Echthrus reluctator*, Cryptinae), and one on the preceding branch (in Labeninae). This character is largely associated with woodboring hosts with the exception of some taxa within Labeninae.

## Discussion

### The wood-boring syndrome

Of the eight characters associated with wood boring hosts, we have identified at least six that constitute the wood-boring syndrome: (1) *first valvula (1vv) with teeth*; (2) *elongated terebra*; (3) *elongated abdominal tergum 9*; (4) *modified apical flagellomere*; (5) *rugulose mesoscutum*; and (6) *first valvula enclosing second valvula*. We exclude the other two characters–ovipositor guides and modified ventral margin of clypeus–from the wood-boring syndrome. The former is an autapomorphy, while the latter is ambiguously coded (see below for details). While we have identified these characters as part of the syndrome, there are likely a few others that were not captured in the Bennet et al. [32] dataset, such as the striation on the ovipositor sheaths or the presence of a steering mechanism (see Table 1 for more details).

In the following sections, we conduct an in-depth analysis of all eight characters, highlighting the limitations identified in Bennett et al.'s [32] coding and suggesting future research directions. Characters identified as part of the syndrome are marked with an asterisk (*) for clarity. The remaining two characters, that we are not including within the syndrome, are discussed without any symbol.

**\*First valvula (1vv) with teeth.** Based on the literature (Table 1), teeth on the first valvulae are necessary for drilling to reach the host. However, our results suggest their presence is not exclusive to wood-boring taxa (Fig 2). This trait may precede the development of the long terebra and the host shift, emerging four times before the switch to a wood-boring host. Thus, the presence of teeth may serve as an early adaptation facilitating the evolution of a wood-boring lifestyle.

In Bennett et al.'s [32] original coding, the teeth on 1vv are coded as a presence/absence character. This coding worked here as we found a strong correlation between the teeth on 1vv and the wood-boring host shift. However, we also recorded the characters in other non-wood-boring parasitoids (e.g., Pimplinae), suggesting the idea that the presence of teeth on the terebra in Hymenoptera can be used for different purposes other than ovipositing into a woodboring host. For example, Ass and Funtikow [104] highlighted how some of the teeth in basal Hymenoptera may serve the purpose of sawdust removal during probing, while Fritzén and Sääksjärvi [105] reported that *Clistopyga* Gravenhorst, 1829 (Pimplinae) teeth on the 1vv are used to cling to the spider host if it attempts to escape.

**\*Elongated terebra.** An elongated terebra is highly convergent within Ichneumonidae, having evolved independently at least 13 times (Fig 2). Our results highlight a very strong correlation between the wood-boring substrate and the presence of an elongated terebra, which co-occurs five times with the host shift. Clearly the elongated terebra provides an important function to address the unique challenges posed by wood as a host substrate. However, exceptions such as the genus *Agonocryptus* (Cryptinae), which lacks a long terebra, exist. Additionally, numerous non-wood-boring taxa possess an elongated terebra, such as *Stethantyx* and *Tersilochus* (Tersilochinae), indicating a broader functionality.

While a long terebra is undoubtedly crucial for reaching hosts within wood, it is also necessary for reaching any kind of host which is concealed within its substrates (e.g., hosts within fruits or galls). More importantly, even when the host is in close proximity to the substrate surface, and thus not deeply concealed, a long ovipositor may be essential for locating the host. In some Hymenoptera, indirect mechanisms employed to steer the terebra for locating hosts have been previously documented [e.g., 37,77,78], typically associated with taxa possessing a long terebra (e.g., Banchinae, Glyptini).

In Bennett et al.'s [32] original coding, the terebra is coded as discrete binary character (longer than the metasoma or not). This coding worked here as we recovered a significant correlation of a long terebra with wood-boring parasitism. However, a continuous quantitative measurement would better assess the full spectrum of variation in terebra length across Ichneumonidae. Dissections would also improve the measurements to better capture the length of the entire ovipositor, rather than just the extruded portion. Further, dissections would allow for assessment of the rotation of the ovipositor capsule which may impact the measurement. For example, when the rotation of the capsule is ~90° (e.g., many Pimplinae), the bulb (anterior area of the second valvula) is exposed and the entire length can be measured. However, when little to no rotation occurs, the anterior area of the second valvula (2vv) remains hidden, and the measurement would underestimate the length.

**\*Elongated abdominal tergum 9.** Our results show that an elongated abdominal tergum 9 is exclusively associated with wood-boring (evolved independently five times), and in most lineages (4/5), it appears to evolve after the host switch. This contrasts with Santos and Perrard

[23], who did not find a significant correlation between this character and the wood-boring substrate in Cryptinae.

Bennett et al. [32] coded the dimension of abdominal tergum 9 as a multistate character, coding together the elongation of the tergite with the presence/absence of "horn and bosses". We adjusted the coding for our analysis (see S1 File). Future analyses could involve the separate coding for the presence or absence of horns or bosses to assess if these characters are exclusively correlated with an elongated T9 (and therefore to a woodboring host shift) or if they are present more broadly across Ichneumonidae. Further, the length could be a quantitative character to provide a more refined understanding of the evolution of this trait.

A recent study suggested the enlargement of these apical segments may relate to muscle strength during oviposition [106]. However, there are no muscles external to the ovipositor capsule directly involved in the oviposition process. As elucidated by various authors [e.g., 107,108], only one muscle connects T9 with the 1st valvifer, and it does not significantly impact strength during oviposition. The remaining 11 muscles are located internally within the ovipositor, either alternately moving the two pairs of valvifer or indirectly manipulating the valvulae (and consequently, the terebra) [107–109]. Thus, the enlarged apical segments could likely serve a different function, potentially related to sensory or mechanical activities, aiding in maneuvering, or providing tactile feedback for the elongated terebra.

*__Modified apical flagellomere.__ Our results highlight a moderate correlation between the hammer-like flattened apical flagellomere and the wood-boring lifestyle. This character is predominantly associated with wood-boring parasitism (Fig 2), though one species, *Baryceros texanus* (Cryptinae), does not target wood-boring hosts but still must locate a concealed host (within the non-lignified plant tissue). Thus, this antennal modification is not exclusive to the wood-boring lifestyle, but certainly is important for deeply concealed host location.

In Bennett et al.'s [32] original coding, the modified apical flagellomere is a multistate character with states for a hammer-like modification and another state for apical projections. We transformed this character to be binary with either having a hammer-like modification of the segment or not (see S1 File). Certainly other modifications may be useful for finding hosts as noted by Broad and Quicke [52]. A comprehensive comparative anatomy study of the entire antennae in Ichneumonidae would be helpful for understanding all modes of host detection that parasitoid wasps utilize, like the presence of sensilla at the tip (for a comprehensive review of their morphology, refer to Beutel et al. [110]). For instance, Rhyssinae relies on vibrational rather than chemical cues [111]. Studies conducted on Cynipidae (Hymenoptera, Cynipoidea) [112], Encyrtidae (Hymenoptera, Chalcidoidea) [113], and Braconidae (Hymenoptera, Ichneumonoidea) [114] showed diverse sensillae functions that could serve as a baseline for similar studies in Ichneumonidae.

*__Rugulose mesoscutum.__ A mesoscutum with strong rugulose dorsal sculpture is found exclusively in taxa associated with a woodboring lifestyle, but having evolved independently only twice within Ichneumonidae (Fig 2). In Bennett et al.'s [32] original coding, the character is coded as discrete binary that captures whether or not the dorsal sculpture of mesoscutum has transverse rugae. This coding worked here as we recovered a significant correlation of a rugulose mesoscutum with wood-boring parasitism. However, future research should evaluate the skeleto-musculature of the mesosoma in Ichneumonidae to determine if muscle and skeletal organization influence the observed variations in mesoscutum surface sculpture. Internal morphological analysis is indispensable for evaluating seemingly identical external traits [e.g., 11,12,115].

*__First valvula (1vv) enclosing second valvula (2vv).__ Differently from Santos and Perrard [23], our results highlight a strong correlation between the 1vv enclosing 2vv and the wood-boring lifestyle. This character evolved four times (Fig 2), once in a non-woodboring clade

(e.g., *Labium*). In Bennett et al.'s [32] original coding, the character is coded as a discrete binary character that captures whether or not the 1vv encloses the 2vv was sufficient for this study. Santos and Perrard [23] suggested that this character helps stabilize the terebra during probing. Another possibility is that it contributes to the actual drilling process by enhancing the surface area, possibly allowing for an expansion of the number of teeth or a more ideal placement for drilling. Understanding the functionality of this character would be helpful to understand its evolution, which will likely require a detailed analysis of the components of the second valvula that expand into a lobe, the number, placement, and direction of teeth, and the overall organization of the terebra. Quicke et al. [85] conducted transverse sections across nearly all subfamilies of Ichneumonidae, revealing a remarkable diversity of modifications within terebra organization. However, this work could be expanded on through investigations on taxa with a dorsal lobe, such as Pimplinae and Labeninae.

**Ovipositor guides.** In Bennett et al.'s [32] original coding, the character is coded as discrete binary character that captures whether or not there are ovipositor guides. The moderate correlation between the ovipositor guides and the wood-boring lifestyle we recovered (Fig 2) was surprising given that this character appears autapomorphic for Rhyssinae [32,116]. The need for ovipositor guides is likely necessary to brace the extremely long terebra of Rhyssinae during oviposition into a hard substrate, as noted by observational data [71,82]. This character is also functionally convergent with other strategies used to maneuver and brace a long ovipositor to drill into wood. For instance, members of the subfamily Labeninae possess a groove either proximo-ventrally on the metasoma or medially on the hind coxae to guide the ovipositor [19,68]. Similarly, some Cryptinae (e.g., genus *Mesostenus* Gravenhorst, 1829) support the long ovipositor by means of a groove in the hind femora [117]. Thus, functionally equivalent traits may need to be considered in addition to trait similarity for a more comprehensive understanding of host-driven syndromes in parasitic wasps [3,118]. Further, understanding the specific muscles within the metasoma may reveal further convergences across other wood-boring taxa that are less visible through just external morphological evaluation.

**Modified ventral margin of clypeus.** Our results highlight a very strong correlation between a wood-boring lifestyle and the development of tooth-like processes (denticles) or the reduction of the clypeus to a concave, bilobed structure (see S1 File). However, as the reduction and denticles are combined here for binary coding, we think we cannot adequately capture this character's association with a woodboring host and thus refrain from drawing any definitive conclusions. When this character is mapped onto the phylogeny (Fig 2), it is clear that there are instances where these clypeal modifications are present without a corresponding association with oviposition within wood. This is notably observed in genera such as *Mastrus* (Cryptinae), *Erythrodolius* (Sisyrostolinae), and *Collyria* (Collyriinae), which attack concealed but not woodboring larvae. Consequently, it is possible that the modifications in the clypeus represent a convergent evolution among wasps emerging from hosts and exiting substrates, rather than being exclusively linked to wood-boring oviposition.

Future research should focus on the reduction of the clypeus rather than the modifications of the ventral margin. In fact, in the sister family Braconidae, taxa attacking wood-boring hosts either have a concave clypeus and labrum, which form a depression behind the mandible (cyclostome Braconidae), or the clypeus appears reduced without a depressed labrum (e.g. in Helconinae, a non-cyclostome). Both these modifications are thought to enhance the range of movement of the mandible during the gnawing process required for emerging from the pupal chamber [56]. Therefore, a comparative anatomy study on the oral cavity of Ichneumonoidea could provide potential characters associated both with feeding behavior and emergence from the host as highlighted in other taxa by various authors [e.g., 57,119,120]. For the moment, given the uncertainty, we preferred to avoid including this character into the woodboring syndrome.

## Limitations of the dataset

The strength of our study lies in the comprehensive testing of these characters through correlation analyses and Ancestral State Reconstructions (ASR). Correlation analyses provide statistical support, while ASR offers deeper insights into the evolution and functionality of these characters based on existing literature. However, as we utilized characters from an existing dataset originally intended for phylogenetic reconstruction and not syndrome testing, we acknowledge certain limitations. First, not all characters associated with wood-boring parasitism, as detailed in Table 1, were present in the utilized dataset. It is foreseeable that more characters will be included into the woodboring syndrome once they are scored in a more comparative analysis. Second, trait coding was not always adequate to analyze specific traits for syndrome testing, such as the clypeal margin or clypeal reduction that may be associated with woodboring or deeply concealed larvae. It would be ideal to also test traits associated with deeply concealed hosts, not just woodboring hosts to see which traits have a broader functionality for reaching the host beyond the specific substrate. Finally, including trait data from the sister-group Braconidae would likely provide a stronger comparative dataset, enhancing our understanding of the wood-boring syndrome on a broader evolutionary scale.

## Conclusions and future directions

In this study, we investigated the existence of a convergent trait syndrome in Ichneumonidae correlated with a wood-boring lifestyle. Our findings indicate that there are at least six characters involved in the wood boring syndrome, namely *first valvula (1vv) with teeth*, *elongated terebra*; *elongated abdominal tergum 9*, *modified apical flagellomere*, *rugulose mesoscutum*, and the *first valvula enclosing second valvula*. Another character–*ventral margin of the clypeus*–remains uncertain, and we foresee its potential inclusion in the syndrome following further refinement of its coding, specifically to look at clypeal reduction. We also found that the *first valvula enclosing second valvula* is strongly correlated with the woodboring substrate, differently from what Santos and Perrard [23] found, even though its functionality is still uncertain.

This study was limited by the use of an existing morphological dataset that was not specifically designed to test syndromes. While we were able to modify some character coding for purposes of this study, some traits that have been associated with the woodboring lifestyle were not tested in this study. Future studies would benefit from more detailed morphological examinations that involve both internal and external characters. The morphological exploration of Ichneumonidae has been historically limited, with characters often reused without substantial refinement [e.g., 11]. While progress in defining certain characters within Hymenoptera has been evident in recent years (e.g., the *mesopleural sulcus* in de Brito et al. [121]), similar advancements in Ichneumonoidea have been lacking. Myrmecologists have made significant strides in exploring and defining internal skeleto-musculature [e.g., 122,123]. In Ichneumonidae, aside from the terebra, the exploration of head capsule and metasoma remains limited but promises to reveal intriguing characters that could reshape our understanding of the group's evolution, potentially challenging previously assumed evolutionary pathways. Recent studies [e.g., 124], have also underscored the unexplored role of glands in parasitoids, prompting questions not only about their evolutionary significance but also their implications during oviposition. Expanding research efforts in these directions will be crucial for advancing our knowledge of host-driven convergent evolution in Ichneumonidae.

## Supporting information

**S1 File. Select morphological characters from Bennett et al. [1].** We modified some characters to better suit the analysis of the current study, mainly creating binary characters from

multistate characters. We followed Sereno's [2] logical basis, in which there are characters and statements. The characters have 3 components: (1) the primary locator (L1), the entity bearing the quality and that alone cannot unambiguously identify the feature of interest; (2) the secondary locator (L2), the containing structure (not always necessary); and (3) the variable (V), which is the aspect that varies; and (4) the variable qualifier (q) which is the phrase that modifies the variable. The statement has only one component which is the character state (vn) which is the mutually exclusive condition of characters. Below, we use the abbreviation in brackets for each part of the character description to facilitate understanding. A rationale for selecting some of the characters not historically correlated with woodboring are provided below characters 11–17, and 19. Alignment of terminology with the HAO can be found in Supplemental Data S2.
(PDF)

**S2 File. Long branch exclusion tests, summary of results.**
(PDF)

**S3 File. Ancestral State Reconstructions of the nine characters.**
(PDF)

**S1 Table. Anatomical terms used for skeletal features, cross-referenced to an ontological (formal) definition (Hymenoptera Anatomy Ontology; URI = Uniform Resource Identifier).**
(DOCX)

**S2 Table. Chimeric alignment of the taxa between Sharanowski et al.** [1] dataset and Bennett et al. [2].
(DOCX)

**S3 Table. Summary of the chi-square test for best-fitting model for the Ancestral State Reconstructions (ASR) for the woodboring substrate (#1) and the eight characters with a moderate to strong correlation with the wood-boring substrate.** Character # is based on the character matrix in Supplementary Data S1. The shaded boxes indicate the best fitting model for that character. MK1 is state transitions occur at equal rates, Asymm2 is state transitions occur at different rates.
(DOCX)

**S4 Table. Results of the correlation analyses.** (A) Characters vs. substrate (woodboring), with *Rhimphoctona* (Campopleginae) as originally coded as a woodborer; (B) Characters vs. substrate (woodboring), with *Rhimphoctona* (Campopleginae) not coded as a woodborer; (C) Characters vs. character correlated with the substrate (see above), with *Rhimphoctona* (Campopleginae) as originally coded as a woodborer. The two model, dependent and independent are presented with the two runs, and the Bayes factor (BF) for each the two runs is presented. A calculation of the average of the two BF results is also provided. Number preceding the parenthesis refers to the numbering in Supplemental S1, while the one in parenthesis reflects the number of the character in Bennet et al. [1].
(DOCX)

## Acknowledgments

The first author (DDP) would like to express deep gratitude to István Mikó (University of New Hampshire) and Lars Vilhelmsen (Natural History Museum of Denmark) for their invaluable

support over the years. Their guidance and encouragement have been instrumental in shaping the education of an aspiring morphologist.

Article processing charges were provided in part by the UCF College of Graduate Studies Open Access Publishing Fund.

## Author Contributions

**Conceptualization:** Davide Dal Pos, Barbara J. Sharanowski.

**Data curation:** Davide Dal Pos, Barbara J. Sharanowski.

**Formal analysis:** Davide Dal Pos, Barbara J. Sharanowski.

**Funding acquisition:** Barbara J. Sharanowski.

**Investigation:** Davide Dal Pos, Barbara J. Sharanowski.

**Methodology:** Davide Dal Pos, Barbara J. Sharanowski.

**Supervision:** Barbara J. Sharanowski.

**Writing – original draft:** Davide Dal Pos.

**Writing – review & editing:** Davide Dal Pos, Barbara J. Sharanowski.

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
