## [Decision Letter · Decision Letter 0]

6 Aug 2024

PONE-D-24-25452A host driven parasitoid syndrome: convergent evolution of multiple traits associated with woodboring hosts in Ichneumonidae (Hymenoptera, Ichneumonoidea)PLOS ONE

Dear Dr. Dal Pos,

Thank you for submitting your manuscript to PLOS ONE. After careful consideration, we feel that it has merit but does not fully meet PLOS ONE’s publication criteria as it currently stands. Therefore, we invite you to submit a revised version of the manuscript that addresses the points raised during the review process.

1. the results and discussion part need some improvement. Kindly verify the duplication of the sentences or statements.2. the inclusion part should be revised in accordance to the reviewer comments. 

3. The graphical parts should be more clarified. 

4. look for the typographical and grammatical errors. 

5. Overall the manuscript is nicely written in good manner. The manuscript can be fit for publication after correction

We look forward to receiving your revised manuscript.

Kind regards,

Salman Khan, Ph.D.

Academic Editor

PLOS ONE

2. In your manuscript, please provide additional information regarding the specimens used in your study. Ensure that you have reported human remain specimen numbers and complete repository information, including museum name and geographic location. 

For more information on PLOS ONE's requirements for paleontology and archeology research, see https://journals.plos.org/plosone/s/submission-guidelines#loc-paleontology-and-archaeology-research.

Reviewers' comments:

Reviewer's Responses to Questions

**Comments to the Author**

1. Is the manuscript technically sound, and do the data support the conclusions?

Reviewer #1: Yes

Reviewer #2: Yes

2. Has the statistical analysis been performed appropriately and rigorously? 

Reviewer #1: Yes

Reviewer #2: Yes

3. Have the authors made all data underlying the findings in their manuscript fully available?

Reviewer #1: Yes

Reviewer #2: Yes

4. Is the manuscript presented in an intelligible fashion and written in standard English?

Reviewer #1: Yes

Reviewer #2: Yes

5. Review Comments to the Author

Reviewer #1: I have gone through the manuscript entitled, "A host driven parasitoid syndrome: convergent evolution of multiple traits associated with woodboring hosts in Ichneumonidae (Hymenoptera, Ichneumonoidea)."

My specific comments are as under:

1. Ms is prepared systematically with all essential elements of long literature based paper with essential statistical analysis.

2. Results drawn are substantial and suitable for reaching to conclusion.

3. Literature review carried out thoroughly and included properly.

4. Duplication of some points is there in results, discussion and conclusion. After retaining essential points, extraneous matter should be eliminated in order to enrich the Ms.

5. Some extraneous matter should be deleted/ dropped from the part "Disenssion"

Recommendation:

Ms is strongly recommended for publication in PLOS ONE after minor revision given in the above points (4&5

Reviewer #2: Review Comments:

1. Detail in Methods: Expand on the phylogenetic methods and the criteria for selecting and analyzing traits.

2. Figures and Tables: Ensure all visual aids are clear, well-labeled, and support the text effectively.

3. Proofreading: Conduct a thorough proofreading to catch any minor errors or inconsistencies.

Overall, the manuscript presents a valuable study on the convergent evolution of traits in parasitoid wasps, with well-supported findings and significant implications for evolutionary biology.

6. PLOS authors have the option to publish the peer review history of their article (what does this mean?). If published, this will include your full peer review and any attached files.

Reviewer #1: No

Reviewer #2: **Yes: **Dr Mohsin Ikram

---

## [Author Response · Author response to Decision Letter 0]

23 Aug 2024

We included this text also as doc file:

Dear Editor,

We sincerely thank you and the reviewers for your time and effort in evaluating our manuscript. We greatly appreciate the constructive feedback provided, which has helped us refine and improve our work.

We have carefully addressed all the comments and made the necessary revisions to the manuscript. Below is a summary of the changes made:

1. Phylogeny and Dataset:

 o Reviewer Comment: "The use of a newly constructed phylogeny and an existing morphological dataset is appropriate for the study. However, more details on the phylogenetic methods and the dataset would be 

 beneficial for reproducibility."

 o Response: We acknowledge the reviewer's concern regarding the need for more details to ensure reproducibility. However, we have some difficulties in understanding how to improve this part as there are no specific 

 comments that allow us to understand where deficiencies may lie. This is what we provided in the original text, which we feel provides more than sufficient data for reproducibility:

 1.Provided comprehensive information on the datasets used.

 2. Detailed the process of combining the two datasets from previously published works, including a supplementary file with listed chimeric sequences (S3 Table).

 3. Specified the tool used for analyzing the phylogenetic tree, including model selection and nodal support.

 4. Included the results of the exclusion tests, with additional information available in the S1 File.

 5. For the Bayes Traits and Ancestral State Reconstruction analyses, we specified, the model tests, parameters, and priors and included the transition rates that best fit each character in Supplemental Table S4 . 

 6. We also specified and cited all software used in the above analyses and the tool used to annotate the phylogeny with morphological characters.

 Unless any specific requests are provided, we feel strongly that the above six points are enough to help reproducibility and comprehension of the overall process that took us to build the phylogeny 

2. Trait Analysis:

 o Reviewer Comment: "The identification of six traits correlated with the wood-boring lifestyle is a key part of the methodology. It would be helpful to have a more detailed description of how these traits were selected 

 and analyzed."

 o Response: We have revised this section to include the following explanation: " We selected 20 characters from the141 coded by Bennett et al. [32] for analysis. The first character is the oviposition substrate itself, as 

 in lignified tissue or not (Character 1, Supplementary Data S1). Then, we conducted a thorough review of the literature and identified 10 of Bennett et al.'s [32] characters that have been historically associated with 

 a woodboring lifestyle (see Table 1 and characters 2-10, and 18, Supplementary Data S1). Then, we chose eight additional characters not previously documented but potentially relevant for ovipositing into wood, 

 based on their overall shape and body position. A rationale for these choices is provided in S1 File (characters 11-17, and 19). This approach was based on the premise that not all traits related to a wood-boring 

 lifestyle may have been previously identified. All other characters from Bennett et al.'s [32] study were clearly unrelated to oviposition or emergence from wood (e.g., all larval characters) and were thus excluded."

We hope this detailed description meets the reviewer's expectations.

Additionally, we have re-checked the entire bibliography, correcting formatting issues caused by an error in the EndNote coding of the PlosOne template. While most corrections were automatically updated by the system, all references are now accurately formatted. We also updated the file names of the Supplementary material.

We hope these revisions satisfactorily address the reviewers' comments, and we look forward to the next steps in the review process.

Thank you once again for your valuable feedback and support.

Best regards,

The Authors

---

## [Decision Letter · Decision Letter 1]

13 Sep 2024

A host driven parasitoid syndrome: convergent evolution of multiple traits associated with woodboring hosts in Ichneumonidae (Hymenoptera, Ichneumonoidea)

PONE-D-24-25452R1

Dear Dr. Dal Pos,

We’re pleased to inform you that your manuscript has been judged scientifically suitable for publication and will be formally accepted for publication once it meets all outstanding technical requirements.

Kind regards,

Salman Khan, Ph.D.

Academic Editor

PLOS ONE

Additional Editor Comments (optional):

The authors have done all the comments prescribed by the reviewer. The manuscript is found suitable for publication in Plos One. 

Reviewers' comments:

Reviewer's Responses to Questions

**Comments to the Author**

1. If the authors have adequately addressed your comments raised in a previous round of review and you feel that this manuscript is now acceptable for publication, you may indicate that here to bypass the “Comments to the Author” section, enter your conflict of interest statement in the “Confidential to Editor” section, and submit your "Accept" recommendation.

Reviewer #1: All comments have been addressed

Reviewer #2: All comments have been addressed

2. Is the manuscript technically sound, and do the data support the conclusions?

Reviewer #1: Yes

Reviewer #2: Yes

3. Has the statistical analysis been performed appropriately and rigorously? 

Reviewer #1: Yes

Reviewer #2: N/A

4. Have the authors made all data underlying the findings in their manuscript fully available?

Reviewer #1: Yes

Reviewer #2: Yes

5. Is the manuscript presented in an intelligible fashion and written in standard English?

Reviewer #1: Yes

Reviewer #2: Yes

6. Review Comments to the Author

Reviewer #1: Manuscript has been prepared nicely with essential elements. the whole paper is based on literature with some addiotional characters. data has been analysed systematically and results have been drawn properly.

Reviewer #2: Authors have incorporated all suggested corrections and authors made all data underlying the findings in their manuscript fully available. So, I will recommend to accept this manuscript for publication.

Thank You

7. PLOS authors have the option to publish the peer review history of their article (what does this mean?). If published, this will include your full peer review and any attached files.

Reviewer #1: No

Reviewer #2: **Yes: **Dr. Mohsin Ikram

---

## [Editor Report · Acceptance letter]

20 Sep 2024

PONE-D-24-25452R1 

PLOS ONE

Dear Dr. Dal Pos, 

I'm pleased to inform you that your manuscript has been deemed suitable for publication in PLOS ONE. Congratulations! Your manuscript is now being handed over to our production team.

Kind regards, 

on behalf of

Dr. Salman Khan 

Academic Editor

PLOS ONE